# Spray Flame Synthesis (SFS) of Lithium Lanthanum Zirconate (LLZO) Solid Electrolyte

**DOI:** 10.3390/ma14133472

**Published:** 2021-06-22

**Authors:** Md Yusuf Ali, Hans Orthner, Hartmut Wiggers

**Affiliations:** 1IVG, Institute for Combustion and Gas Dynamics—Reactive Fluids, University of Duisburg-Essen, 47057 Duisburg, Germany; yusuf.ali@uni-due.de (M.Y.A.); hans.orthner@uni-due.de (H.O.); 2CENIDE, Center for Nanointegration Duisburg-Essen, 47057 Duisburg, Germany

**Keywords:** spray-flame reactor, solid electrolyte, Garnet (Li_7_La_3_Zr_2_O_12_), nanoparticles, Li-precursor

## Abstract

A spray-flame reaction step followed by a short 1-h sintering step under O_2_ atmosphere was used to synthesize nanocrystalline cubic Al-doped Li_7_La_3_Zr_2_O_12_ (LLZO). The as-synthesized nanoparticles from spray-flame synthesis consisted of the crystalline La_2_Zr_2_O_7_ (LZO) pyrochlore phase while Li was present on the nanoparticles’ surface as amorphous carbonate. However, a short annealing step was sufficient to obtain phase pure cubic LLZO. To investigate whether the initial mixing of all cations is mandatory for synthesizing nanoparticulate cubic LLZO, we also synthesized Li free LZO and subsequently added different solid Li precursors before the annealing step. The resulting materials were all tetragonal LLZO (*I4_1_/acd*) instead of the intended cubic phase, suggesting that an intimate intermixing of the Li precursor during the spray-flame synthesis is mandatory to form a nanoscale product. Based on these results, we propose a model to describe the spray-flame based synthesis process, considering the precipitation of LZO and the subsequent condensation of lithium carbonate on the particles’ surface.

## 1. Introduction

The constant depletion of conventional fossil fuels and the latest trend of decarbonization has proliferated the interest in alternative energy conversion associated with suitable energy storage systems. In this context, lithium-ion batteries are of increasing interest, both as buffer storage and as a decentralized energy source in electric vehicles. Here, energy density, storage capacity, and (thermal) stability in particular are playing an increasingly important role. Current lithium-ion battery systems—especially in electric vehicles—require complex temperature management [1,2]. Many of the related challenges can be overcome by using solid ionic conductors instead of the flammable liquid electrolytes used so far [3,4], such as inorganic [5], polymer [6], and organic/inorganic composites. Moreover, the manufacturing costs of solid-state batteries can be significantly reduced [6,7]. Regarding energy and power density, Li metal as a high energy density anode can be employed with solid electrolytes (SEs) and in contrast to many liquid electrolytes, SEs are stable under high electrochemical oxidation potentials up to 5.0 V vs Li/Li^+^ [8]. In addition, solid composite electrolytes (SCE) show the possibility to act as a physical separator, reducing the risk of Li dendrite growth [9] and eliminating the need for a mechanical separator used in conventional cells. Inorganic SEs such as garnet type Li_7_La_3_Zr_2_O_12_ (LLZO) exhibit a number of advantageous properties, such as negligible electronic conductivity, high temperature battery performance, large electrochemical stability window, acceptable Li conductivity (~1 mS cm^−1^), and high elastic modulus [10,11,12,13,14]. These properties make LLZO as a one of the most promising SCE that can meet the commercialization criteria [15]. Garnet family ceramic materials were first reported more than a decade ago by Thangadurai et al., and since then, researchers have taken a huge interest in modifying its properties [16]. In 2007, Murugan et al. reported one of the first high room temperature total ionic conductivity of 4.67 × 10^−4^ S cm^−1^ [17]. However, regarding materials synthesis, there is one important challenge due to the fact that LLZO has two polymorphous, a tetragonal phase (*I**4_1_/acd*), which is thermodynamically stable at room temperature but with low ionic conductivity (~10^−6^ S cm^−1^), and a good conducting cubic phase (*Ia**-3d* and *I-43d*) [18,19]. It is quite difficult to get the pure cubic phase due to Li loss during the required high temperature sintering process and unwanted side products the pyrochlore phase La_2_Zr_2_O_7_ (LZO) forms [20,21]. A range of factors, such as the initial Li content [22], synthesis conditions [23,24,25], and sintering temperature/condition/dopants [18,26,27,28], affect the final ionic conductivity of LLZO. The Li depletion can be addressed by taking excess Li precursor or, according to Pfenninger et al., providing a Li reservoir such as Li_3_N. This reservoir was used in a sandwich manner between LLZO, supplying Li during the sintering process, as well as decreasing the sintering temperature from 1000 to 700 °C [29]. Sintering additives, e.g., LiO_2_-B_2_O_3_-SiO_2_-CaO-Al_2_O_3_, help to lower the sintering temperature and to modify grain boundaries [30]. However, the synthesis of highly conductive pure cubic LLZO is still an open challenge.

The cubic phase of LLZO also known as high temperature phase transforms to the tetragonal form at room temperature and heteroatom doping at different crystalline site is a suitable way to stabilize the cubic phase at room temperature. Most of the studies deal with substitutional doping of the lithium sites. The main approach is towards the introduction of about 0.4–0.5 Li vacancies per unit formula, which is necessary to preserve the cubic phase [18,27]. These vacancies can be introduced by adding elements like Aluminum or Gallium. In cases when the amount of the doping material is not sufficient, a mixture of cubic and tetragonal phase has been observed. Stabilization of the cubic phase is also possible by substitutional doping of the Zr site with higher valence ions, such as Ta^5+^ [31], Nb^5+^ [32], Mo^6+^ [33], and Sb^5+^ [34], but also with Bi^3+^ [35,36]. Moreover, doping of the La site with Nd^3+^ [37] has also been investigated, as well as simultaneous multiple doping. All approaches to stabilize the cubic phase aim at creating oxygen vacancies, thus achieving a reduction of lithium ions per formula unit. In the case of our studies, we used the established doping with aluminum, although we did not extensively study the role of Al doping as it is well researched [22,38,39]. Kubicek et al. have also demonstrated that not only cation vacancy, but also oxygen vacancy can alter the total ionic conductivity [40].

Like the other ceramic materials, the synthesis procedure of LLZO follows either wet phase reaction [41] or solid phase reaction [42]. Typically, ball milling (2–5 h) [43] and utilization of organic complexing agents [44], followed by high-temperature (≥1000 °C), time consuming (~24–48 h) multistep sintering [40] is needed to synthesize cubic LLZO in a classical procedure. However, these procedures are not only difficult to scale up (because of several process steps, such as precipitation, filtration, washing, drying, and calcination etc.), but also render micrometer size particles. These are of limited use due to both high sintering temperatures and the production of polymer/ceramic hybrid materials.

Spray flame synthesis is a well-studied method to produce functional nanometer particles, typically with a mono modal particle size distribution [45]. Moreover, the spray flame method is an effective synthesis procedure to produce multi component particles while retaining the same stoichiometry in the precursor solution with different morphology, e.g., hollow, porous, and core-shell, etc. [46]. In the case of spray flame synthesis, the precursors (typically inorganic salts, e.g., nitrates, carbonates, and sulfates, or metalorganics) are generally dissolved in appropriate solvents, delivered to a nozzle, and then atomized by a dispersion gas flow (pure oxygen in our case). The resulting spray droplets are ignited by a pilot flame. Particles are formed due to: (i) the combustion of previously completely evaporated solvents and precursor components (gas-to-particle route where the precursors generally undergo full combustion and CO_2_ and H_2_O form as side products), (ii) the combustion of spray droplets with incomplete evaporation of solvents and precursors (droplet-to-particle route, leading to bigger, typically dense or hollow spherical particles), (iii) the nucleation of condensable species, and (iv) subsequent coagulation and sintering before deposition on a filter [47]. This fast and usually kinetically controlled process technology enables the formation of many phases that are difficult to produce by different methods. Indeed, even doping is easily accessible. Yi. et al. synthesized LLZO nanoparticles in a flame process and fabricated sintered thin films thereof to measure the ionic conductivity [48]. The authors achieved almost completely dense and crystalline LLZO thin films with a density of about 94% of the theoretical value and a good ionic conductivity of ~0.2 mS cm^−1^). However, the authors did not discuss and explain the synthesis process used, which was based on costly, metal organic compounds. Although the above-mentioned spray-flame synthesis method has many advantages, it also has its own drawbacks, e.g., the sample size distribution can be broad and bimodal in the case of a mixed gas-to-particle and droplet-to-particle particle formation process, as described before. At high precursor loading levels, large aggregates can be formed, leading to low tap densities, which might limit the further use of the materials or require further mechanical processing such as grinding. Moreover, due to some incomplete combustion, the particle surface may absorb organic residues that will hinder its applicability or require additional annealing steps. 

In this paper, we have investigated step wise the gas-phase synthesis of nanoparticles with subsequent calcination to produce phase-pure cubic LLZO. We investigated the dependency between precursor solution present in the spray-flame reaction and the final phase and how it can be altered after the calcination step. Moreover, we propose a model to support the formation of an amorphous, Li-containing layer on the surface of the as synthesized nanoparticles, especially compared to mechanical mixing. 

## 2. Experimental Section

Different solutions for three separate experiments were produced in order to investigate a range of different formation and processing procedures towards the synthesis of LLZO. Depending on the initial solution composition, they are denoted by Sol_LLZO_, Sol_LZO_, and Sol_LO_ (Table 1). Sol_LLZO_ consisted of all elements which are necessary to form cubic LLZO. Sol_LZO_ contains the same composition but no lithium salt, while the Sol_LO_ contains lithium nitrate only. 

Lithium nitrate (LiNO_3_ × xH_2_O, Alfa Aesar) and aluminum nitrate (Al(NO_3_)_3_ × 9H_2_O, Merck) were used as the precursor for Li and Al respectively. Lanthanum acetate (La(OAC)_3_ × xH_2_O, Sigma-Aldrich), and zirconium-tetra-propoxide in 70% propanol solution (Zr(C_3_H_7_O)_4_, Sigma Aldrich) were used as the precursor for La and Zr respectively. As lanthanum-salts have poor solubility in alcohols, lanthanum acetate was dissolved in propionic acid while all other precursor components were dissolved in iso-propanol. The two mixtures were then mixed. Thus, the resulting mixture consisted of a mixture of propanol and propionic acid (1:1 by volume). 

At first, amounts of the precursors (Sol_LLZO1_) were mixed proportional to the intended ratio of the elements Li, La, and Zr in the target material (Li_7_La_3_Zr_2_O_12_). In a second mixture (Sol_LLZO2_), based on literature data [49], aluminum (in a ratio of 1:3 compared to lanthanum) was used as a stabilizer for the cubic form of LLZO. Moreover, due to unsatisfactory results and after intensive literature research, the lithium content was also increased by 50% to compensate for the loss of lithium during synthesis and subsequent annealing. 

A pictorial diagram of the spray flame reactor (manufactured in the university’s own workshop) to process the different solutions is presented in Figure 1. The reactor consists of a spray nozzle which is placed inside a vessel made of stainless steel. All gases were fed to the reactor by mass flow controllers (Bronkhorst High-Tech BV, Ruurlo, The Netherlands). A photo of the spray flame taken during the synthesis of LLZO can be found in Appendix A.

The freshly prepared precursor mixtures were introduced to the reactor by a syringe pump. The flow rate of the liquid was set to 2 mL/min. The liquid was dispersed inside the reactor by a two-fluid nozzle with oxygen as dispersion gas (6 slm). Details of the spray flame reactor and the two-fluid-nozzle nozzle have been already published elsewhere [50]. To ignite the spray, a premixed CH_4_/O_2_ flame served as a pilot flame. The flows of methane and oxygen were 1 and 2 slm respectively. The combination of pilot and spray flame typically reaches maximum temperatures around 2500 K [50], which is more than sufficient for complete decomposition of the precursors. An additional flow of 380 L/min air (coaxial sheath gas) was fed to the reactor to minimize particle deposition at the reactor walls and to cool down the combustion gases before entering the filter unit. The sheath gas also lowers the relative humidity in the reactor off gas to avoid condensation. Finally, the produced particles were deposited on a filter sheet and collected for further processing and characterization. The particle collection time was in the range of 1–2 h. After the consumption of all precursor liquid, the gas flows were stopped, the reactor flushed with air and the filter apparatus was unclosed. Around 2 g of synthesized material were collected each run. For the calcination experiments the as-synthesized materials were put into alumina crucibles and heated inside a horizontal tube furnace (Carbotherm). During heat treatment (1000 °C, 1 h), the tube furnace was kept under oxygen atmosphere to oxidize carbonaceous by-products.

## 3. Characterization

Powder diffraction XRD patterns were measured using an X-ray diffractometer (Empyrean diffractometer PANalytical with Cu Kα radiation, from Malvern, UK). FTIR spectra were measured using attenuated total reflection mode with an FTIR spectrophotometer (Bruker Vertex 80, Bruker Optik GmbH, Ettlingen, Germany), equipped with a beam splitter, a DigiTect DLaTGS detector, and an infrared source. Transmission electron microscopy (TEM, Jeol JEM-2200FS, Japan Electron Optics Laboratory Company, Tokyo, Japan) was used for particle morphology, size, and structure determination. The surface areas of the as-synthesized powders were measured using a Brunauer–Emmett–Teller (BET) device (Nova 2200 from Quantachrome, Boynton Beach, FL, USA, now 3P Instruments GmbH). XPS spectra were recorded using a VersaProbe II (ULVAC-PHI, Chanhassen, MN, USA) equipment with Al Kα radiation. Thermogravimetric analyses (TGA/DSC) were carried out with a Netsch STA 449 F1 (NETZSCH-Gerätebau GmbH, Selb, Germany) under air with a heating rate of 10 k/min up to 1200 °C combined with gas analysis by quadrupole mass spectrometry (QMS 403 D, NETZSCH-Gerätebau GmbH, Selb, Germany). 

## 4. Results and Discussion

To analyze the crystal structure of as synthesized and annealed materials, the powders prepared from Sol_LLZO1_ were examined with X-ray diffraction. Figure 2a shows the respective XRD pattern of the as synthesized nanoparticles. It clearly indicates an almost complete match with the cubic pyrochlore phase La_2_Zr_2_O_7_ (LZO, ICSD 253063, space group Fd-3m) [51]. Thus, it must be concluded that lithium has not incorporated in the crystal structure of the as-synthesized nanoparticles. Since any indication of a second crystalline phase, e.g., Li_2_CO_3_ as detected by Djenadic et al., is missing [52], it is expected that a lithium-containing species is present in amorphous form. This is also supported by more detailed XRD investigation presented later.

Based on Li 1s and C 1s XPS measurements (Appendix A), it can be concluded that lithium is present on the particles’ surface as Li_2_CO_3_ with some traces of lithium oxide. When annealing the as-synthesized powder (LLZO1, 1000 °C, 1 h, oxygen atmosphere), the sample turned into the tetragonal LLZO phase (space group I41/acd, Figure 2b). Thus, tetragonal LLZO has formed via a solid-state reaction from crystalline LZO and amorphous Li_2_CO_3_. Some evaporation of lithium during heat treatment or the absence of an agent stabilizing the cubic phase (e.g., Al doping) could be the reason for obtaining the tetragonal instead of the desired cubic phase. 

To further validate this claim and to synthesize the intended cubic phase, we have prepared a solution with 50% excess Li and minor amounts of Al (NO_3_)_3_ as a dopant (designated as Sol_LLZO2_). Similar to the product obtained from Sol_LLZO1_, the as-synthesized material still shows the pyrochlore phase of LZO (Figure 2c). Again, any indication of additional crystalline phases containing lithium or aluminum is missing. However, in contrast to the material from Sol_LLZO1_, the XRD pattern of the annealed powder clearly indicated that the sample corresponds to the desired Li^+^ conductive cubic phase (Figure 2d, space group Ia3d) [27]. 

From the fact that the pristine material produced has the pyrochlore structure in all cases, it can be deduced that the addition of a lithium precursor during spray flame synthesis is not necessary but can be added in a second step before annealing. To prove this, we synthesized a material where the precursor solution (designated as Sol_LZO_) was prepared without any Li source. As expected, the XRD of the respective product is almost identical to that of the materials obtained from Sol_LLZO1_ and Sol_LLZO2_. Thus, regarding the formation of pyrochlore LZO, there is no difference in the nucleation process whether lithium is added or not. 

The particle morphology, size, and size distribution of as-synthesized LLZO2 and LZO1 were determined using TEM (Figure 3). LLZO2 seems to be more agglomerated than LZO1 (Figure 3a,d). In both cases, the insets of the HRTEM images (Figure 3b,e) show the single crystalline structure of LZO with lattice fringe distances of the pyrochlore phase (0.32 nm) corresponding to the (222) plane when compared with the XRD data. For the corresponding electron diffraction pattern, please refer to Appendix A.

The obtained size distributions were plotted in histograms and fitted with a log-normal function. The average particle size for LLZO2 and LZO1 are 9.2 and 9.4 nm, respectively, with a geometric standard deviation of 1.3 and 1.7. These results do not agree particularly well with those of the BET measurements. The measured specific surface areas of 22 and 32 m^2^/g, respectively, correspond to an average size of about 46 and 31 nm (assuming monodisperse, spherical particles). The reason for the difference to the TEM results is the existence of few larger, spherical particles up to the micrometer range, which are not formed via the gas-to-particle process but via the droplet-to-particle process (Appendix A). Even if their proportion is not high in terms of numbers, their comparatively large mass has a significant effect on the measured BET surface area. The low magnification image of LLZO2 (Figure 3a) suggests particles with an amorphous matrix or shell, as seen in the highlighted area. The image is clearly different from that with LZO1 particles (Figure 3d), where clear particle and grain boundaries can be seen. We associate this to the fact that amorphous Li species is present on the particles’ surface, most probably carbonate as discussed before. In Figure 3a, we have marked an area where particles are embedded in an amorphous matrix. A high-contrast image of this area is almost impossible. Unlike the crystalline particles, the amorphous material (most probably Li_2_CO_3_ as identified by XPS, X-ray photoelectron spectroscopy) consists only of very light elements with deliver low contrast. Moreover, it was not possible to obtain high-quality TEM images with high resolution since they changed during the examination. We attribute this to the fact that the amorphous layer decomposes when irradiated with the electron beam, which leads to blurred images and is an additional indication of the presence of amorphous regions.

The lower BET area for LLZO2 is also in line with the assumption that LLZO2 consists of the pyrochlore phase particles covered with an amorphous lithium phase. However, it can be concluded that the particle formation process is not affected by the presence of the lithium precursor. Rather, the results suggest that lithium carbonate precipitates on the freshly synthesized pyrochlore nanoparticles. This subsequent precipitation also explains why the lithiated phase is not directly accessible via the spray flame synthesis route. All other constituents are homogeneously distributed within the particles as can be seen from TEM elemental mapping of La, Zr, and Al in LLZO2 and LZO1 (Appendix A, respectively). 

Based on the results, the following growth model is proposed for this spray flame synthesis as shown in Figure 4. Initially, the vaporized precursors decompose to release the metals, followed by the nucleation of La_2_Zr_2_O_7_. We assume that, in this process, lithium is not incorporated, because lithium—similar to the formation process of zinc oxide from flame synthesis [53]—intermediately forms gaseous lithium hydroxide (boiling point of LiOH: 924 °C). As a result, LLZO cannot be formed either, because lithium continues to be bound in gaseous form and is therefore not available. Only in the further course and with decreasing temperature is Li_2_CO_3_ formed in the CO_2_-containing atmosphere of the reactor off gas, which then condenses on the already formed particles of LZO.

To investigate whether the initial addition of lithium is mandatory regarding the synthesis of LLZO, three different precursor materials were mixed with Li-free LZO and investigated: commercial LiNO_3_, Li_2_CO_3_, and a lithium compound prepared by spray-flame synthesis from a solution of LiNO_3_ in propanol and propionic acid (designated as Sol_LO_), keeping all reaction parameters the same as during LLZO synthesis (see Figure 5), which is described more detailed below. Sol_LO_ was processed similarly to the other solutions (Sol_LLZO1_ and Sol_LLZO2_) before, but in contrast, a crystalline Li-material was obtained consisting of mainly lithium propionate and Li_2_CO_3_ (Figure 5b). This product is further referred to as LO. XRD fitting of LO shows Li-propionate (PDF: 00-037-1558, space group: P21/c, Monoclinic cells) as a main phase (70.8%) and some Li_2_CO_3_ (ICSD: 69133, space group: C 1 2/c 1) (29.1%) as a secondary phase. However, the Li_2_CO_3_ signals are extremely broadened, indicating very low crystallinity. 

Prior to annealing, the various mixtures were investigated using TGA/DSC/QMS to determine the extent to which the previously used temperature of 1000 °C is sufficient for conversion. The corresponding measurement of LLZO2 served as a comparison. 

An initial mass loss (8%) for LLZO2 (Figure 6a) between ambient temperature and <230 °C is ascribed to physi- and chemisorbed H_2_O removal from the sample as no release of CO_2_ and NO_x_ was observed with QMS. The subsequent mass loss (13%) between 290 and 390 °C is assigned to removal/oxidation of residual organic precursor components and unburned solvent. This mass loss corresponds to a sharp exothermic signal (~350 °C, see inset in Figure 6a) signifying the organic residues are oxidized in air releasing CO_2_ and H_2_O [48]. These residues mainly arise from remaining acetate groups from the precursors and carboxylate groups from the solvent used in the spray-flame synthesis [54]. The following mass loss (~13%) between 700 and 950 °C coupled with endothermic change in the sample (see inset) can be attributed to the melting and decomposition of Li_2_CO_3_ releasing CO_2_ [55]. Most probably, during this decomposition process, Li species melts and incorporates into the crystal structure of LZO to form the intended cubic LLZO. Thus, considering a heating rate of 10 °C min^−1^, the TGA measurements can explain that annealing at 1000 °C for one hour is sufficient to form c-LLZO. As expected, the temperature behavior of LZO (Figure 6a) is almost identical to that of LLZO up to 700 °C. However, since it does not contain any lithium, the corresponding decomposition from 700 °C is naturally absent.

The thermal analysis of LO (Figure 6b) indicates mainly three step mass losses at similar temperatures as observed for LLZO2. The first step can be associated with the release of moisture, followed by an exothermic decomposition/oxidation of the propionate (identified by XRD) around 370 °C The third step between 720 and 970 °C corresponds to the CO_2_ release from lithium carbonate decomposition. The high mass loss indicates that the material mainly consists of Li_2_CO_3_. 

To analyze the temperature-dependent behavior of the physical mixtures of LZO1 with the different lithium sources, TGA studies were also performed. As expected from the results shown before, after an initial release of water (where the water loss for the mixture with LiNO_3_ is particularly high due to its water of crystallization), decomposition/oxidation of residual organics from LZO1 is observed in all cases around 350 °C (Figure 6c). The subsequent thermal decomposition of the nitrate starts at around 600 °C while LZO + Li_2_CO_3_ and LZO + LO show the almost similar TGA characteristics due to the fact that both lithium sources mainly contain Li_2_CO_3_. From the TGA analysis, it can be concluded that for all mixtures annealing at 1000 °C for one hour should be sufficient to produce LLZO.

Before annealing, the powders of Li precursors were mixed with LZO1, intensively mortared, and compressed into pellets with 5 mm diameter. Then, the pressed pellets were heated under O_2_ flow at 1000 °C for 1 h, which was identified to be sufficient to synthesize the LLZO phase. For longer residence times, (two and three hours, respectively), the LLZO phase decomposes back to the pyrochlore phase due to the increasing evaporation of lithium (see Appendix A). The XRD measurements of the mixed powders as well as the sintered pellets are shown in Figure 7. The diffraction patterns of the initial mixtures show the LZO phase and Li_2_CO_3_ (Figure 7a, compare also Figure 5a). Any specific LiNO_3_ or LO peak could not be observed because these lithium precursors have no or only very low crystallinity compared to LZO1. Surprisingly, despite the fact that LZO1 contains the same amount of aluminum to stabilize the cubic structure, the XRD patterns of all the three calcined powder mixtures (Figure 7 b) match not with the cubic, but with the tetragonal LLZO phase, with minor impurities of lithium carbonate in the material produced from LZO1 + Li_2_CO_3_, understandable given the fact that lithium carbonate decomposes at somewhat higher temperatures than LO and LiNO_3_ (compare Figure 6c). 

These results lead to the conclusion that a simple physical mixing of the powders in combination with a comparatively short annealing step is not sufficient to obtain the desired cubic structure. We expect that longer annealing at higher temperatures would result in the desired material, but this would be accompanied by an unwanted strong crystal growth. Therefore, in order to actually produce a nanocrystalline and cubic lithium lanthanum zirconate, the extremely good “mixing” of lanthanum zirconate and Li precursor, as obviously occurs during the condensation of Li_2_CO_3_ on the fresh particle surface, seems to be inevitable. 

## 5. Conclusions

Here we have successfully synthesized nanocrystalline cubic lithium lanthanum zirconate (LLZO) based on a spray-flame process followed by a short thermal treatment. In this case, the addition of aluminum as a dopant is mandatory to stabilize the cubic structure. Our experiments showed that direct synthesis of LLZO in the spray flame reactor is not possible. To explain this, a model was developed that assumes rapid nucleation of lanthanum zirconate (LZO) while the Li-containing species initially remains in the gas phase and only condenses on the surface of the freshly formed particles as a result of cooling and reaction to Li_2_CO_3_ downstream of the flame zone. Comparative studies in which a nanocrystalline, lithium-free LZO was produced and only then mechanically mixed with different Li precursors showed that the short sintering times were not sufficient to produce the desired cubic crystal structure of the LLZO. All products exhibited the undesirable, poorly ion-conducting tetragonal phase. Thus, the particularly good mixing of LZO and Li precursor, as occurs in spray flame synthesis, shows particular advantages here. To our knowledge, this is the only bottom-up method to produce this nanomaterial.

## Figures and Tables

**Figure 1 materials-14-03472-f001:**
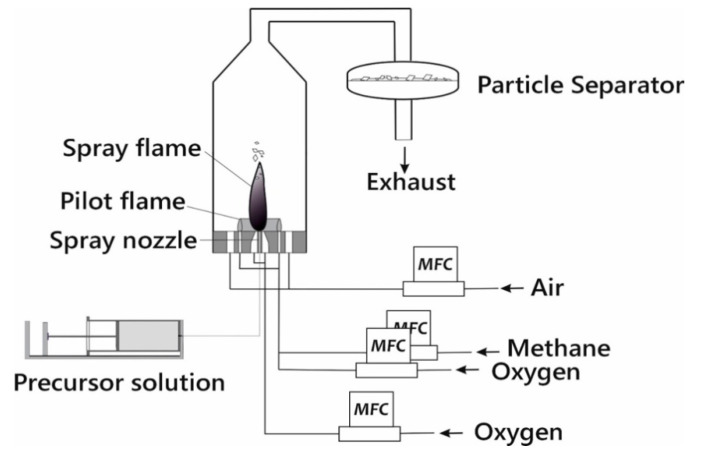
Graphical illustration of the spray-flame reactor employed to synthesize LLZO1, LLZO2, LZO1 and LO.

**Figure 2 materials-14-03472-f002:**
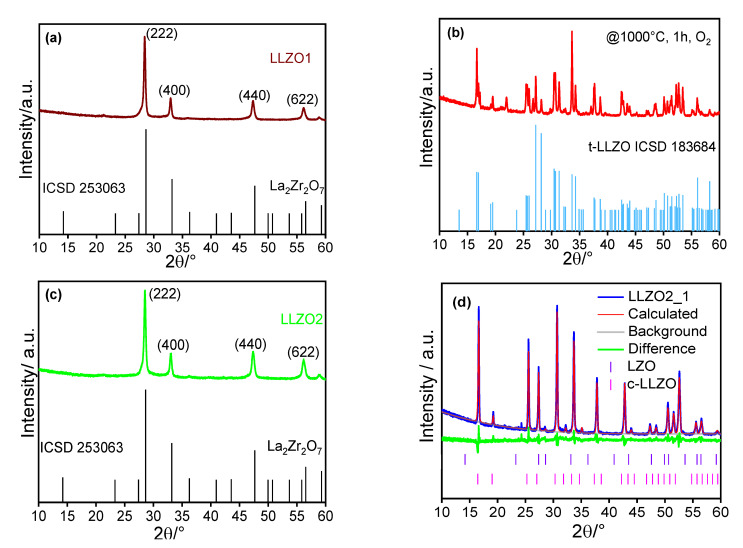
X-ray diffraction pattern of as-synthesized nanoparticles from Sol_LLZO1_ (**a**) and XRD of the calcined product (annealed at 1000 °C, 1 h, O_2_ atmosphere) shown in (**b**). XRD of as-synthesized nanoparticle from Sol_LLZO2_ and its calcined product LLZO2_1 (annealed at 1000 °C, 1 h, O_2_ atmosphere) shown in (**c**,**d**) respectively.

**Figure 3 materials-14-03472-f003:**
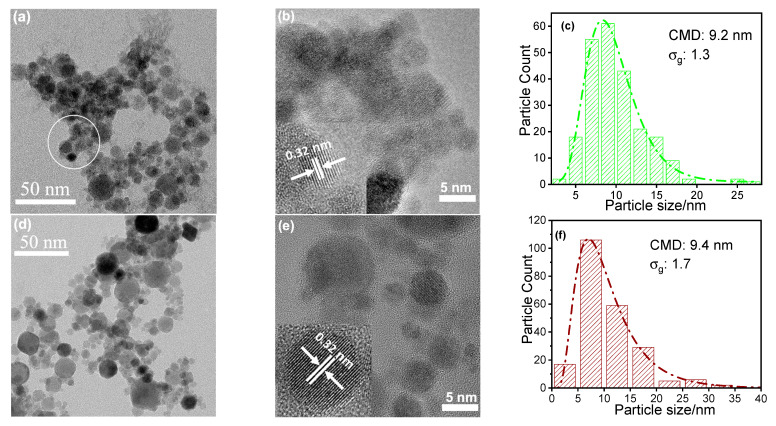
TEM (**a**), HRTEM (High Resolution Transmission Electron Microscopy) (**b**), and particle size distribution (**c**) of LLZO2. TEM (**d**), HRTEM (**e**), and particle size distribution (**f**) of LZO1.

**Figure 4 materials-14-03472-f004:**
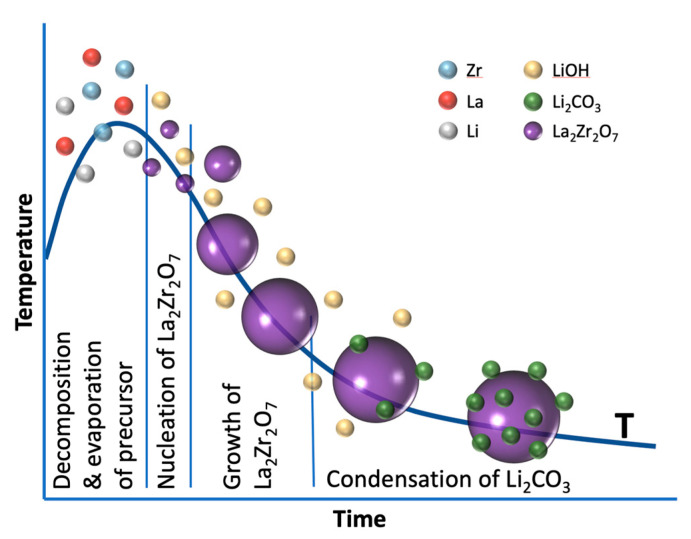
Scheme describing the formation of La_2_Zr_2_O_7_ with subsequent condensation of Li_2_CO_3_ downstream.

**Figure 5 materials-14-03472-f005:**
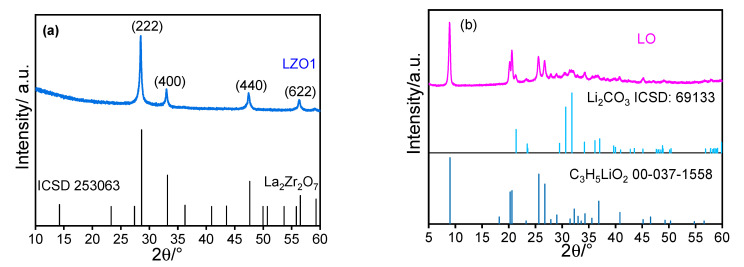
X-ray diffraction pattern of as-synthesized LZO from Sol_LZO_ (**a**,**b**) of the product from burning the solution Sol_LO_ (LO), indicating a mixture of Li propionate (PDF: 00-037-1558) and Li_2_CO_3_ (ICSD: 69133).

**Figure 6 materials-14-03472-f006:**
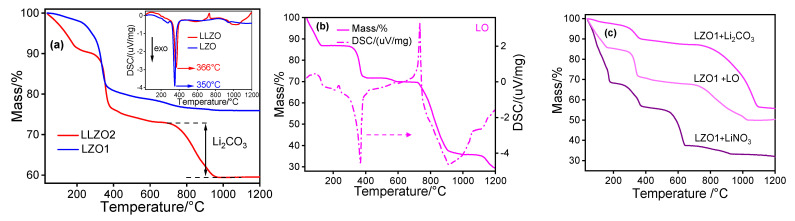
(**a**) TGA analysis of LLZO2 and LZO1 and the respective DSC measurements (see inset). (**b**) TGA and DSC analysis of LO. (**c**) TGA of LZO1 physically mixed with LiNO_3_, Li_2_CO_3_ and LO (from spray flame synthesis), respectively.

**Figure 7 materials-14-03472-f007:**
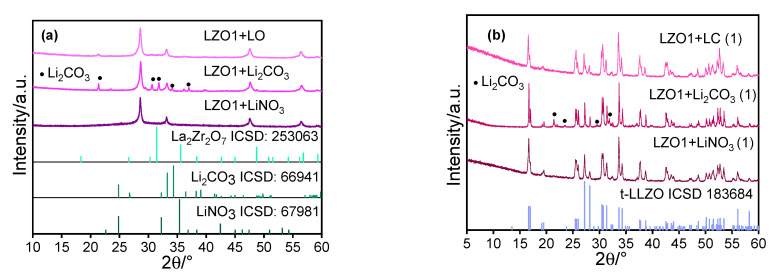
XRD of the mixture of LZO and different Li precursors (**a**) and of the calcined products (**b**) presented together with the diffraction pattern of tetragonal LLZO.

**Table 1 materials-14-03472-t001:** Precursor solution chart and their corresponding nomenclature.

Nomenclature	Product Name	Precursor				Solvent
		LiNO_3_	La(OAC)_3_	Al(NO_3_)_3_	Zr-Propoxide	Propanol + propionic acid 1:1 by vol
Sol_LLZO1_	LLZO1	✓	✓	ⅹ	✓	✓
Sol_LLZO2_	LLZO2	(50%excess) ✓	✓	✓	✓	✓
Sol_LZO_	LZO	ⅹ	✓	✓	✓	✓
Sol_LO_	LO	✓	ⅹ	ⅹ	ⅹ	✓

## Data Availability

Not applicable.

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
