# Peer review of "Spray Flame Synthesis (SFS) of Lithium Lanthanum Zirconate (LLZO) Solid Electrolyte"

_materials, 2021, doi:10.3390/ma14133472_

Round 1
Reviewer 1 Report
The manuscript titled, “Spray Flame Synthesis (SFS) of Lithium Lanthanum Zirconate (LLZO) Solid Electrolyte”, is interesting study. My comments are below;
- There are many grammatical formatting mistakes, missing references, improper punctuations and so on.
- The introduction part requires modification in terms of up-to-date literature in similar research field.
- Illustration of method has been given, I suggest to add real pictures, also of samples (aq. suspensions), if possible.
- The comparative part of nanocrystalline analysis, other aspects than sintering times and cubic crystal structure.
- Any limitations of spray process could be added.
Author Response
We are grateful for the reviewers' many comments on improving the quality of our work and have carefully revised our manuscript accordingly. Our responses to the reviewers' comments are marked in red.
Reviewer 1
The manuscript titled, “Spray Flame Synthesis (SFS) of Lithium Lanthanum Zirconate (LLZO) Solid Electrolyte”, is interesting study. My comments are below;
- There are many grammatical formatting mistakes, missing references, improper punctuations and so on.
We revised the manuscript accordingly
- The introduction part requires modification in terms of up-to-date literature in similar research field.
The introduction was revised by us, changes are marked in red
- Illustration of method has been given, I suggest to add real pictures, also of samples (aq. suspensions), if possible.
The starting materials are present as clear solutions, the products are inconspicuous powders, photos are not very informative here. A photo showing the reactor in operation was added to the supplementary information (Figure S1).
- The comparative part of nanocrystalline analysis, other aspects than sintering times and cubic crystal structure.
Fig 6a shows the TGA of sample LLZO2 investigated under air atmosphere at the rate of 10 °C/min. The decomposition of LLZO2 ends around 1000°C, so 60 mins is enough to change the pyrochlore phase to the cubic/tetragonal LLZO phase. It shows the efficiency of our synthesis system, the respective information is added to the manuscript.
- Any limitations of spray process could be added.
With regard to the limitations of spray flame synthesis, we have supplemented the introduction accordingly.
Reviewer 2 Report
This is a synthesis paper describing the preparation of lithium lanthanum zirconate using spray flame process. The approach is interesting but there are some major issues that would need further clarification before the publication. The major and minor comments are listed in the attached document.

Author Response
We are grateful for the reviewers' many comments on improving the quality of our work and have carefully revised our manuscript accordingly. Our responses to the reviewers' comments are marked in red.
Reviewer 2
This is a synthesis paper describing the preparation of lithium lanthanum zirconate using spray flame process.
The approach is interesting but there are some major issues that would need further clarification before the
publication. The major and minor comments are listed below.
General comments: There are issues with the automatic referencing to the figures in the text.
We have checked the automatic referencing, should be ok now.
In the order of the appearance in the text:
Page 2, line 83: Not all inorganic salts but rather nitrates are being used in the flame processes, since they will produce gaseous nitrogen oxides when combusted.
We appreciate the advice and have added nitrates as well as sulfates and carbonates as they all essentially just produce gaseous byproducts.
Page 2, line 86: In the flame synthesis the organometallic precursors typically decompose, and the organic compounds undergo complete combustion, forming carbon dioxide and water vapor. The metallic or semimetallic components of the precursor will then nucleate and condensate to form primary particles of metal oxides or metals.
However, if the combustion is not complete for some reason, the particles may form through droplet-to-particle process, which involves evaporation of the solvents.
This is an important hint, and the respective information is now added on page 2 and 3.
Page 2, line 92: What is meant by ~94 % density?
It is the density of the crystalline thin film (compared to the theoretical density of completely dense LLZO). The sentence has been reworded for clarity.
Page 4, line 141: I recommend change the wording, since “the synthesis time” refers to the compound formation, which happens within very short time in the flame and will not take hours. The operation or particle collection time is probably hours.
We are thankful for this hint and have changed the text as suggested
Page 4, line 144: The calcination temperatures and times should be added to the text.
Has been added as suggested
Page 5, Figure 2: Please, change the caption or modify the figure, so that the caption will proceed in the order: a, b, c, d.
The figure caption has been changed as suggested
Page 5, line 203: The authors state that the amorphous coated nanoparticles are clearly visible. However, the amorphous coating cannot be observed in these images. Thus, higher magnification images are required to confirm this statement.
We have replaced the TEM images (Figures 3a and d) with those with a slightly higher resolution and revised the respective description. However, in case of materials prepared with lithium precursor, it was not possible to obtain high-quality TEM images with high resolution, since they changed during the examination. We attribute this to the fact that the amorphous layer decomposes when being irradiated with the electron beam, which leads to blurred images and is an additional indication of the presence of amorphous regions. In Figure 3a, we marked an area suggesting that particles are embedded in an amorphous layer and explained it in the manuscript.
Page 5, line 209: It is stated that single crystalline LZO with lattice corresponding to the (311) plane is observed. However, this cannot be stated based on the images provided and the result should be confirmed from the electron diffraction patterns.
We are very grateful for this note, as we have noticed that the cited reference is obviously incorrect. The mesh spacing of 0.32 nm belongs to the (222) plane of LZO, which is consistent with our XRD studies. In addition, as suggested, we have included an image of the electron diffraction pattern (Figure 4c) whose intensity distribution agrees with that from the X-ray analysis.
Page 6, Figure 3: The images have different magnifications, so the direct comparison of the images is difficult. I recommend providing the images with the same magnifications.
We have changed the magnifications for better comparison, see also comment above.
Page 6, line 233-241 and page 7, line 253 and 265:
It seems that the combustion is not complete in the described process. The unburned organics (page 7, line 253) and precursor residues (page 7, line 265) indicate that flame temperature is rather low, which may be due to the alcohols and/or high dilution. Thus, the residence time in the high temperature is not adequate to cause LiNO3 to decompose. I recommend include the information of the adiabatic temperatures of the system and the dilution ratio, and discussion about their effect on the combustion process and especially the temperatures.
Other alternatives are to try different solvents, which will increase the flame temperature, and/or decrease the dilution to provide longer time in the high temperature to complete the reaction.
Based on additional experiments addressing this topic (not presented in the manuscript so far), we found that LiNO3 decomposition is not the main hurdle here. To support this statement, we added the TGA analyses of precursors to the supplementary (Figure S5). The decomposition of solid LiNO3 happens before 600°C. Moreover, we investigated the thermal decomposition of a dried precursor solution indicating that the thermal decomposition of the Lanthanum precursor (Lanthanum acetate) seems to be one limiting factor. The main unburned organics that have been analyzed for materials made in similar experiments mainly revealed acetate groups from the precursors and carboxylate groups from the solvent. Regarding the temperature profile, we refer to Lit. 48, where the spray flame and also its temperature field is analyzed in detail, indicating maximum flame temperatures around 2500 K. The respective information has been added to the manuscript.
Page 7, line 248: Also provide the chemical formula of “the lithium compound prepared by sprayflame”.
The respective information was added
Page 7, line 252: It should be explained why there is formation of crystalline Li2CO3 when the synthesis is carried out just containing lithium precursors but not when all components are present.
As now discussed in relation to Figure 5, the intensity of the peaks corresponding to Li2CO3 is very weak and the signals are clearly broadened. Thus, it is to be expected that these weak signals are lost in the background noise in the presence of highly crystalline lanthanum zirconate, which is now explained. It is also mentioned in the paper that also in case of physically mixing LZO1 with LO the signals for the lithium species are not visible due to very low crystallinity compared to LZO1.
Page 8, line 284: Li2CO3 has been reported to undergo decomposition around 1300 °C. Thus, the temperature range reported here seem to be rather low. This should be explained in the text.
As shown by Beyer et. al. [1] the decomposition of pure Li2CO3 can be start around 700°C until 950°C which is well matched with our TGA analysis as presented in Figure 6.
Page 8, line 301 and page 9, line 310:
Why this temperature and residence time was chosen? What will happen if the residence is longer?
Longer residence times leads to the decomposition of LLZO due to the release of lithium. After 3h of heating the sample at 1000 °C it almost turns back to pyrochlore phase. Respective results are added to the supplementary information and referred to in the manuscript.
- Beyer, H., et al., Thermal and electrochemical decomposition of lithium peroxide in non-catalyzed carbon cathodes for Li–air batteries. Physical Chemistry Chemical Physics, 2013. 15(26): p. 11025-11037.
Reviewer 3 Report
In this manuscript the authors present an investigation on the synthesis of cubic Al-doped Li7La3Zr2O12 (LLZO) by Flame-Spray-Pyrolysis combined with sintering under O2 atmosphere. The data provide useful insights into the mechanism of stabilization of the cubic-phase versus the tetragonal-phase. They show that the FSP-step produces La2Zr2O7 pyrochlore phase coated with Li-carbonate. This formation is thermodynamically predisposed to form the cubic LLZO under a short thermal treatment. This an interesting research on LLZO synthesis that can be published in Molecules Journal.
Before this however I consider that crucial data are missing. These should eb incorporated in the revised manuscript as follows.
[1] The authors found that Al is required to stabilize the cubic LLZO phase. However the role of Al is poorly mapped. The authors should present data on the dependence of the cubic-phase stabilization on Al/La-ratio.
[2] The role of Al, should be discussed in more in detail. This approach is known in solution-synthesis methods using Al or Tantalum.
[3] The phase diagram of the cubic LLZO vs tetragonal LLZO should be include din the analysis, in relation to reason for the formation of the cubic LLZO by the post-FSP calcination vs. the mixing approach.
Author Response
We are grateful for the reviewers' many comments on improving the quality of our work and have carefully revised our manuscript accordingly. Our responses to the reviewers' comments are marked in red.
Reviewer 3
In this manuscript the authors present an investigation on the synthesis of cubic Al-doped Li7La3Zr2O12 (LLZO) by Flame-Spray-Pyrolysis combined with sintering under O2 atmosphere. The data provide useful insights into the mechanism of stabilization of the cubic-phase versus the tetragonal-phase. They show that the FSP-step produces La2Zr2O7 pyrochlore phase coated with Li-carbonate. This formation is thermodynamically predisposed to form the cubic LLZO under a short thermal treatment. This an interesting research on LLZO synthesis that can be published in Molecules Journal.
Before this however I consider that crucial data are missing. These should be incorporated in the revised manuscript as follows.
[1] The authors found that Al is required to stabilize the cubic LLZO phase. However, the role of Al is poorly mapped. The authors should present data on the dependence of the cubic-phase stabilization on Al/La-ratio.
[2] The role of Al, should be discussed in more in detail. This approach is known in solution-synthesis methods using Al or Tantalum.
[3] The phase diagram of the cubic LLZO vs tetragonal LLZO should be included in the analysis, in relation to reason for the formation of the cubic LLZO by the post-FSP calcination vs. the mixing approach.
Because the reviewers’ questions all deal with more or less one topic, we would like to answer them all together:
Several studies have elucidated the effect of dopant on the phase stability of LLZO. The cubic phase also known as high temperature phase transforms to the tetragonal form at room temperature. As reported in literature, an amount of 0.4 to 0.5 Li vacancies per unit formula is necessary to preserve the cubic phase. These vacancies can be introduced by adding elements like Aluminum or Gallium, which substitutes on the Li-site. In cases when the amount of the doping material is not sufficient a mixture of cubic and tetragonal phase has been observed. Other elements like Tantalum and Niobium substitutes at the Zr site which also stabilizes the cubic phase. Although, we did not extensively study the role of Al doping as it is well researched. The respective information is added to the introduction of our manuscript.
Unfortunately, due to the complexity of the system containing five elements (Al-Li-La-Zr-O), a complete phase diagram of LLZO is still not available.

Round 2
Reviewer 1 Report
I have no further comments to add.
Reviewer 2 Report
The authors have responded to the comments adequately and revised the manuscript accordingly. Thus, I recommend the paper to be accepted for publication.
Reviewer 3 Report
In the revised manuscript the authors provide text improvemnts that comply with my original comments. I suggest acceptanc eof this ms in its current form